# Immobilization of Strontium Aluminate into Recycled Polycarbonate Plastics towards an Afterglow and Photochromic Smart Window

**DOI:** 10.3390/polym15010119

**Published:** 2022-12-28

**Authors:** Mohamed E. El-Hefnawy, Ali I. Ismail, Sultan Alhayyani, Soha T. Al-Goul, Mohamed M. Zayed, Manal Abou Taleb

**Affiliations:** 1Department of Chemistry, Rabigh College of Sciences and Arts, King Abdulaziz University, Jeddah 21589, Saudi Arabia; 2Department of Chemistry, Faculty of Science, Tanta University, Tanta 31527, Egypt; 3Department of Chemistry, Faculty of Science, The Hashemite University, P.O. Box 150459, Zarqa 13115, Jordan; 4Environmental and Occupational Medicine Department, National Research Centre, Giza 12622, Egypt; 5Department of Chemistry, College of Science and Humanities, Prince Sattam Bin Abdulaziz University, Al-Kharj 11942, Saudi Arabia; 6Department of Polymer Chemistry, National Center for Radiation Research and Technology (NCRRT), Egyptian Atomic Energy Authority, Nasr City, Cairo 11762, Egypt

**Keywords:** recycled polycarbonate, strontium aluminate nanoparticles, smart window, afterglow, photochromism

## Abstract

A transparent smart window made of recycled polycarbonate plastic (PCP) waste was prepared and immobilized with strontium aluminate phosphor nanoparticles (SAPN). It has afterglow emission, super-hydrophobicity, durability, photostability, good mechanical properties, ultraviolet protection, and high optical transmittance. To create an afterglow emission polycarbonate smart window (SAPN@PCP), recycled polycarbonate waste was integrated with various concentrations of SAPN (15–52 nm). SAP micro-scale powder was made using the solid-state high temperature method. The SAP nanoparticles were produced using the top-down method. To create a colorless plastic bulk, recycled polycarbonate waste was inserted into a hot bath. This colorless plastic was thoroughly combined with SAPN and cast to create an afterglow luminous smart window. To investigate its photoluminescence properties, spectrum profiles of excitation and emission were measured. According to the luminescence parameters, the phosphorescent colorless polycarbonate plates displayed a change in color to strong green under UV illumination and greenish-yellow in a dark box. The afterglow polycarbonate smart window displayed two emission peaks at 496 and 526 nm, and an absorption wavelength of 373 nm. Upon increasing the SAPN ratio, the hydrophobic activity, hardness, photostability, and UV protection were improved. Luminescent polycarbonate substrates with lower SAPN ratio demonstrated rapid and reversible fluorescence under UV light, while the higher SAPN content in the luminous polycarbonate substrates showed afterglow.

## 1. Introduction

Researchers have attempted to develop home energy-saving products that maximize light transmission while consuming the least amount of energy [1]. Large amounts of glasses have been utilized to create building windows. However, glass windows are very delicate, with safety and financial considerations [2]. Polycarbonate is a large class of polymers with a carbonate repeating unit. Polycarbonates have been used in various fields, such as the reinforcement of automobile tires, conveyor belts, sails, and canoes. They have been used to create holograms, insulators, capacitors, filtration systems, and dielectric films with excellent mechanical performance [3]. Resistance to ageing, heat, water, and chemicals are all characteristics of polycarbonate. Owing to their low cost, mechanical toughness, shatterproof qualities, poor heat conductivity, low shrinking characteristics after curing, and good optical transmittance, polycarbonates has recently garnered a lot of attention [3,4,5]. As a result, polycarbonate presents a valuable material for the creation of smart window. Nonetheless, only a small number of research papers have discussed polycarbonate-based photochromic and afterglow materials with energy-saving capabilities. Smart materials can be defined as products that react to certain stimuli such as light and/or heat, in a reversible and controllable manner [6,7,8].

Smart materials are employed in the creation of protection goods owing to their ability to respond to potentially harmful external stimuli such as extremely high temperatures and hazardous substances [9]. Some smart materials either continue to emit light after switching off the illumination source, showing afterglow emission, or emit light beneath the excitation source to show fluorescence. Fluorescence can only be activated by an excitation source, and when that source is removed, the material light emission stope immediately [10,11,12]. Light-stimulated material may continue to emit light for a few seconds, showing phosphorescence, or for hours, showing afterglow emission. Photochromism is the term for a light-stimulated change in color that takes place in a material when it is exposed to an illumination source and then regains to its origin color when the illumination source is moved away [13]. For smart applications, including electronic displays, ophthalmic lenses, sensors, and packaging, photochromism has proven a fascinating technique. The understanding of photochromism has benefited military camouflage, ultraviolet protection, security barcodes, and smart textiles [14,15]. Most materials that have UV-induced photochromism are colored using organic colorants. However, the limitations of UV-induced chromic organic pigments include poor photostability, photochromism inhibition, and expensive cost, which restricts their applicability [16,17]. The organic pigments photochromism relies on have structural switching. Thus, their photochromic activity is frequently inhibited by steric effects brought on by immobilization in bulk materials. Additionally, continued exposure to UV light may cause organic colorants to gradually photo-degrade, resulting in poor photostability [18,19]. However, no steric effects occur with photochromic inorganic colorants. As a result, photochromic inorganic colorants have enhanced photochromic properties and strong photostability. Due to their excellent photostability, strontium aluminate phosphors (SAPs) have better resistance to fatigue. SAPs are recyclable, non-radioactive, and non-toxic, among other luminous lanthanide-doped inorganic pigments they have great brightness, strong afterglow emission (>10 h), and excellent resistance to water, heat, light and chemicals [10,20,21]. The ratio of SAPs in the material is necessary for determining the target application, including photochromism or afterglow. SAPs have therefore been preferred for use in a variety of smart products [22,23]. Alkaline earth aluminate is incorporated into a polycarbonate matrix as a novel technique for enhancing smart windows with a transparent appearance, photochromism and afterglow. These windows have an energy-saving nature, a tough surface, high optical transmittance, are inexpensive to produce, and highly durable.

In this context, a simple method was employed to produce an afterglow transparent SAPN@PCP with UV protection and hydrophobic properties. SAPN was integrated into polycarbonate (PCP) sheets to make a transparent smart window that emits light. SAPN (average diameter 15–52 nm) was created in the form of nanoparticles and analyzed using a transmission electron microscope (TEM). To enhance the SAP dispersion in PCP, SAPN must be uniformly disseminated in the PCP matrix without agglomeration for the best development of the colorless window. X-ray fluorescence (XRF), electron scanning microscope (SEM), X-ray diffraction (XRD), and energy-dispersion X-ray (EDX) analysis were applied to study the morphology of SAPN@PCP. The manufactured luminous SAPN@PCP morphology, chemical composition, hardness, and hydrophobicity characteristics were examined. To investigate the afterglow characteristics of the created PCP sheets, the excitation and emission spectra were investigated.

## 2. Experimental

### 2.1. Materials

Recycled polycarbonate plastic (PCP) was obtained from the local marketplace (Aquahgroup, Cairo, Egypt). Transparent polycarbonate from plastic bottles was employed in the current study. The molecular weight of polycarbonate plastic bottles typically ranges between 50 k–300 kg mol^−1^. The SAP was prepared using a high-temperature solid-state approach [24] using aluminium oxide (Merck; Egypt), strontium carbonate (Merck; Egypt), boric acid (Merck; Egypt), europium oxide (Aldrich; Egypt), and dysprosium oxide (Aldrich; Egypt). A top-down route [25] was used on the created SAP micropowder in an ES80 Triple Roll Mill to develop the SAPN.

### 2.2. Preparation of Polycarbonate Bars

The PCP waste was firstly shredded and then placed into a hot bath (110 °C) to remove the colored substrates to provide a clear plastic. At 110 °C with mechanical stirring, the generated plastic was thoroughly admixed with SAPN till the admixture turned translucent, indicating the absence of SAPN aggregations. To create the PCP substrates, various SAPN ratios (0, 0.5, 1, 2, 4, 6, 8, 10, 12, 14 and 16% *w*/*w*) were used. The resulting plates were denoted by codes from PCP-0 to PCP-16, respectively. To produce the matching PCP sheet with a 2 cm in width, 6 cm in length, and 0.5 cm in thickness, the produced melts (110 °C) were inserted into sheet-like Teflon plates and allowed to cool for 12 h under ambient conditions. A schematic illustration of the methods involved in making the luminous clear PCP plates is shown in Figure 1.

### 2.3. Characterization Methods

The particle size and shapes of SAPNs were examined by JEOL 1230 TEM (JEOL Co. Ltd., Tokyo, Japan). The XRD pattern of SAPN was determined by a Bruker Advance D-8 (Bruker, Rheinstetten, Germany) with copper radiation (Kα) PCP sheets (1.5406 Å) were examined by SEM (Quanta FEG250; Czech Republic). The chemical content of SAPN@PCP was examined by EDX attached to the SEM. The chemical contents were also investigated by XRF advanced Sequential Axios. The JASCO FP8300 spectrophotometer (Tokyo, Japan) was used to investigate the afterglow spectra of SAPN@PCP. Five minutes of UV exposure were employed to assess the photoluminescent reversibility of PCP sheets. The SAPN@PCP sheet was left in the dark for the following 60 min so that it regained its origin color. The emission spectrum was collected both before and after ultraviolet light was used to excite the sample. By calculating the ultraviolet protection factor (UPF) of the luminescent PCP sheets, it was discovered that they possess UV-protective qualities [26]. The UPF was calculated using a UV-Vis spectrophotometer with the AATCC 183(2010)UVA method. The hardness of the prepared SAPN@PCP sheets was evaluated using a Shore D device (Otto Wolpert-Werke, Ludwigshafen/Rhein, GMBH, Germany). The contact angles [27] of the polycarbonate SAPN@PCP films were analyzed with an OCA15EC device (Dataphysics, Filderstadt, GmbH, Germany).

## 3. Results and Discussion

### 3.1. Preparation of Polycarbonate Window

The high-temperature solid-state preparation of the SAP microparticles was followed by exposure to the top-down method [24] to produce SAPN. According to TEM, the sizes of the particles were between 15 and 52 nm (Figure 2). Figure 2d shows an HRTEM image taken from the edge of lanthanide-doped strontium aluminate particle. To keep the matrix transparent, the use of nanoparticles has proven crucial [28]. As a result, the transparency of the sheet matrix is preserved by the employment of SAP nanoparticles. Using various concentrations of SAPN trapped in the polycarbonate matrix, sheets of colorless polycarbonate with afterglow and photochromic characteristics were created. The luminescence spectra proved that the transparent photoluminescent PCP sheets changed their color to greenish when placed under ultraviolet light. The current photoluminescent polycarbonate sheets can be used for a variety of purposes, such as safety signs, military camouflage, antireflective coatings, smart windows, electricity-free illumination, and secure prints.

The XRD pattern of SAPN and its standard spectrum are shown in Figure 3. The diffraction peaks of SAPN were identical to the pure monoclinic phase of SrAl_2_O_4_ (JCPDS card number 01-074-0794) [24]. No other diffraction peaks were detected, indicating a complete integration of Eu^2+^ and Dy^3+^ in the SrAl_2_O_4_ crystal lattice [24], and showing low temperature monoclinic phase content. The small contents of the lanthanide doping elements had almost no effects on the SrAl_2_O_4_ phase composition.

### 3.2. Morphological Features

Morphologies of the created SAPN@PCP plates are shown in Figure 4 and Figure 5. The elemental compositions (weight percentage) of SAPN@PCP sheets are shown in Table 1 and Figure 6 for three different spots on the polycarbonate plate surface. The SAPN ratio was raised without altering the morphology of SAPN@PCP. The absence of SAP nanoparticles on the surface of the PCP sheets under examination suggests that the SAP nanoparticles were fully encapsulated within the bulk of the PCP. The existence of SAPN in the bulk of the PCP polymer was established by EDX analysis. To show virtually homogeneous dispersion of SAPN in SAPN@PCP, EDX was used at three different points on the polycarbonate surface, which suggests that SAPN was distributed uniformly. Strontium (Sr), oxygen (O), carbon (C), dysprosium (Dy), aluminum (Al) and europium (Eu) were the elements that EDX detected. The polycarbonate hosting bulk (C and O) and SAPN (Al, Dy, Sr, and Eu) represent the main contents of the SAPN@PCP sheets. The prepared SAPN@PCP compositions were also investigated by XRF (Table 2). The EDX method, which is distinguished by a smaller error compared to XRF, can be utilized to establish material chemical content with great accuracy. XRF could potentially be utilized to ascertain the chemical content of a specific item at a concentration greater than 10 ppm. As a result, in comparison to EDX, X-ray fluorescence was only able to determine the elemental compositions of a small number of elements [29]. Al and Sr were both monitored by XRF in the produced SAPN@PCP. Nonetheless, Dy and Eu could not be detected by XRF due to their extremely low molar ratios. The percentages of elements used in the production of luminous sheets and SAPN were discovered to equal the elemental concentrations determined by both methodologies.

### 3.3. Photoluminescence Spectra

The detection of photochromic transition to green was found to be significantly enhanced by the transparent backdrop of SAPN@PCP. In the SAPN@PCP sheets, instantaneous and reversible photochromism was observed. However, the SAPN@PCP films with SAPN ratios equal to or less than 1% showed fluorescence emission with quick reversibility under UV. As a result, when the ultraviolet source was turned off, the fluorescence emission from SAPN@PCP films with a SAPN of 1% or less instantly vanished. The SAPN@PCP sheets, on the other hand, showed delayed reversibility and afterglow emission when the SAPN concentration was more than 1%. Consequently, the SAPN@PCP films with SAPN concentrations more than 1% persisted to emit light even in the dark. Wavelengths at 373 nm (excitation band) and 526 nm (emission band) were recorded for SAPN@PCP. After being exposed to UV light, a weak blue band was observed at 496 nm and a strong green band was observed at 526 nm. The partial negative charge on the carbonate carbonyl oxygen of PCP and the positive charge on Al(III) of SAPN create coordination bonds as a result of the physical entrapment of SAPN inside the bulk of the SAPN@PCP film, and these bonds were found to strengthen the bonding between the polycarbonate chains. Figure 7 and Figure 8 display the excitation and emission spectra of the PCP-12 sheet. The excitation band was detected at 373 nm, demonstrating that the peak intensity strength depended on the SAPN ratio. Strong green emission (526 nm) and mild blue emission (496 nm) were produced by the 4*f*-5*d* transitions of Eu^2+^ [23]. These emission bands can be attributed to two strontium sites in the crystal structure of SrAl_2_O_4_. However, thermal quenching at room temperature could inhibit the blue band. Thus, only an intense green emission could be observed with the naked eye. The luminescence spectra showed no emission bands caused by Eu^3+^, showing that Eu^2+^ was responsible for emission.

UV-stimulated chromism has been used in many anticounterfeiting applications [30]. To make a translucent sheet, PCP and SAPN were mixed. Potent vivid green emission was observed when exposed to UV radiation; however, there were no traces evident when exposed to visible light. This photochromic feature is important in applying printing pattern anticounterfeit technology to a variety of commercial products. The SAPN@PCP sheets can be used for anticounterfeiting purposes, especially for packaging, when a pattern is sliced into rectangular pieces. Straightforward and difficult-to-copy anticounterfeiting patterns can be easily identified by the green emission produced under ultraviolet rays. Therefore, the current method can be viewed as a successful approach to creating a variety of anticounterfeiting materials to enhance marketing, with financial and social benefits. To ensure transparency, the generated SAPN@PCP plates underwent an optical transmittance test. When SAPNs were added to PCP at a high ratio, the optical transmittance decreased. The colorless SAPN@PCP sheet (PCP-1) was observed to exhibit a typical emission at 1% and had the greatest green fluorescence under UV light. The PCP-12 sheet, which exhibited a clear appearance and greener afterglow under UV radiation, was shown to be the most efficient afterglow at a concentration of 12%. PCP-1 and PCP-12 sheets had optical transmittances of 66% and 86%, respectively. The colorless PCP-1 and PCP-12 sheets emitted a green light when exposed to UV radiation. The existing photoluminescent SAPN@PCP films can be easily made invisible to build anticounterfeiting patterns on commercial objects.

### 3.4. Photostability Study

The emission of SAPN has been ascribed to 4f^6^5D^1^↔4f^7^ transition of Eu^2+^ ions. There were no emission peaks seen for either Dy^3+^ or Eu^3+^ to support a full exchange of Eu^3+^ to Eu^2+^. The fundamental function of dysprosium is to promote the creation of light traps, which can then be discharged in the dark to allow Eu^2+^ to return to its ground state [22,23]. The afterglow and photochromic materials need to be fatigue-resistant to offer long-term photostability and endurance. PCP-12 was put through a coloration/decoloration operation before and after ultraviolet illumination for several cycles and displayed great reversibility (Figure 9).

### 3.5. UV-Protection and Hydrophobicity Screening

Smart windows that block UV radiation can aid in reducing the risk of erythema, skin cancer, and sunburn [25]. The UV shielding abilities of the SAPN@PCP sheets were tested using a range of SAPN ratios, as shown in the results presented in Table 3. Due to the strong ultraviolet absorbance of SAPN embedded in PCP, PCP-0.5 has a substantially better UV shielding than PCP-0. As the SAPN content in the PCP matrix increased, the photoluminescent transparent SAPN@PCP sheets had greater UV-blocking. The colorless PCP sheets turned green after exposure to light due to the SAPN phosphor. The photoluminescent SAPN@PCP sheets can be utilized as smart windows. When smart windows with photochromic features are open during the daytime, they are subjected to a lot of ultraviolet light. The smart window absorbs ultraviolet rays and emits green light to limit the amount of UV that enters the building. The SAPN@PCP plate returns to its origin colorless form in the presence of weak sunlight, increasing the amount of light that is present inside. Hydrophobicity was measured using static contact angles, and the results revealed that as the SAPN ratio was increased, material water-repellent activity also increased. When SAPN was embedded in PCP (PCP-0.5), the contact angle of the produced SAPN@PCP was increased to 134.0°. The contact angle increased from 134.0° to 148.3° when the SAPN amount in the PCP matrix increased from PCP-0.5 to PCP-10. The contact angle then increased slightly as the SAPN ratio climbed from PCP-10 (148.3°) to PCP-16 (150.7°), but the roughness decreased as a result of the smaller gaps between the SAP nanoparticles (Table 3).

### 3.6. Hardness Properties

The degree of deformation that impacts material durability is affected by the material hardness. Hence material hardness is a crucial indicator of how well a material performs [25,27]. PCP was used to provide the greatest clearness while keeping a smooth surface. The hardness characteristics showed that the hardness improved as the SAPN ratio increased. The hardness of the luminous SAPN@PCP was assessed at various SAPN amounts (Figure 10). The hardness was observed to decrease as the SAPN ratios increased, then it increased once more to higher values. As the SAPN ratio increased, the hardness decreased from PCP-0 (8.36 kg/mm^2^) to PCP-10 (6.00 kg/mm^2^), and then increased from PCP-10 (6.00 kg/mm^2^) to PCP-16 (6.92 kg/mm^2^). Raising SAPN from PCP-0 to PCP-10 caused the impact to fall from 8.57 to 5.98 MPa. The impact was observed to increase to 6.63 MPa for PCP-16. The reason for this can be attributed to Al(III) in SAP, which may act as a coordinative catalyst between the carbonate carbonyl oxygen in the PCP polymeric chains, improving hardness and impact resistance.

## 4. Conclusions

By mixing strontium aluminate phosphor nanoparticles (SAPN) into recycled polycarbonate waste, colorless plastics with photochromic and afterglow features were created, resulting in smart windows with high photostability, high rigidity, ultraviolet protection, high optical transmittance, durability, and hydrophobicity. Using TEM, the morphology of SAPNs was investigated. The created SAPNs were between 15 and 52 nm in diameter. Numerous techniques were employed to examine the morphology of SAPN@PCP. The sustained afterglow and photochromism of SAPN@PCP, which were transparent (373 nm) in visible light and green (526 nm) light under UV irradiation, were validated by photoluminescence spectra. Contact angles were evaluated to determine improved hydrophobicity due to an increase in SAPN content, increasing from 134.0° to 150.7°. Hardness was also examined, and the results showed that there were no appreciable changes when the SAP nanoparticle ratio was increased. Different SAPN phosphor concentrations were investigated on the SAPN@PCP sheets. Under a UV lamp, it was discovered that the colorless polycarbonate with the optimum greenish fluorescence had an optimal emission at a SAPN amount of 1%. The SAPN@PCP film with the optimum greener afterglow when exposed to UV was observed in the most efficient glow-in-the-dark film at a SAPN amount of 12%. The photoluminescent SAPN@PCP sheets were examined for their high reversibility and photostability.

## Figures and Tables

**Figure 1 polymers-15-00119-f001:**
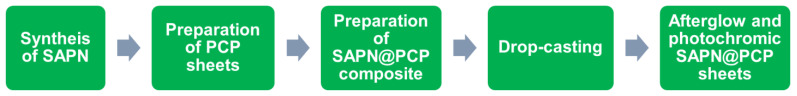
Scheme illustrating the manufacture of PCP films.

**Figure 2 polymers-15-00119-f002:**
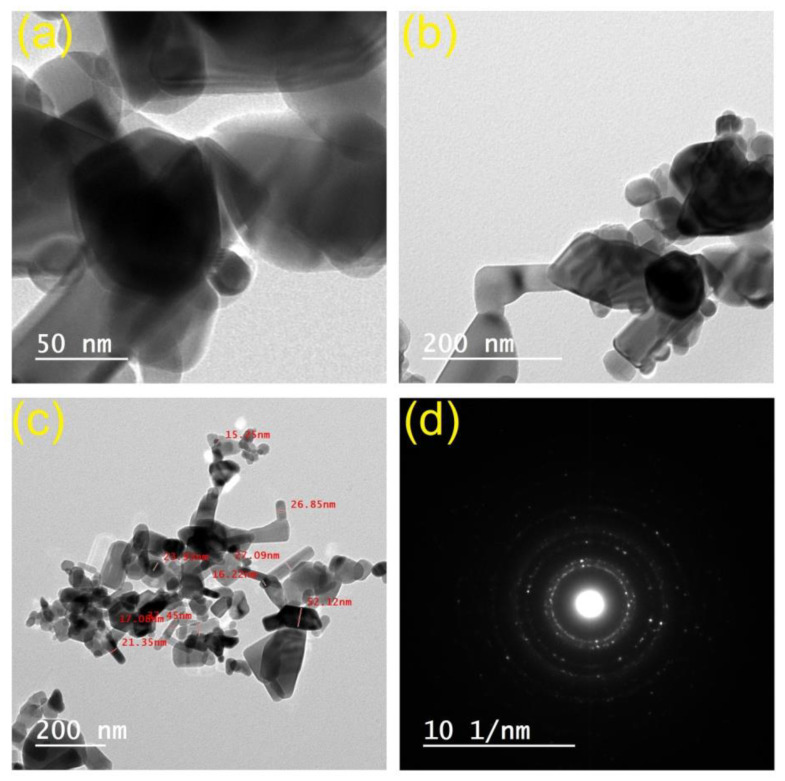
TEM images (**a**–**c**), and selected area (electron) diffraction (**d**) of SAPN.

**Figure 3 polymers-15-00119-f003:**
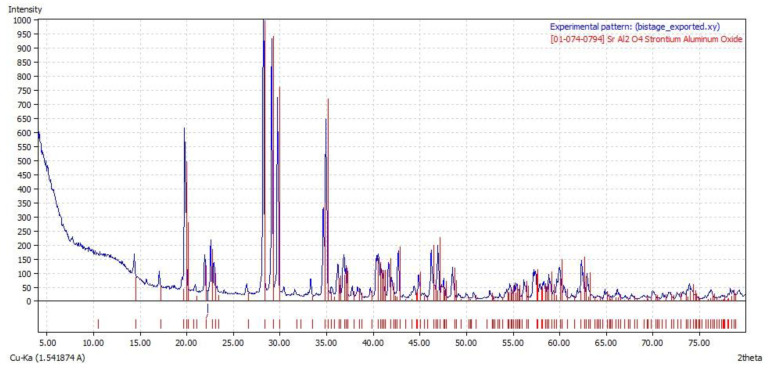
XRD patterns of SAPN (**top**) and its standard spectrum (**bottom**).

**Figure 4 polymers-15-00119-f004:**
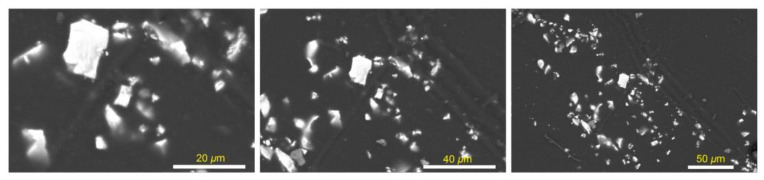
SEM images of PCP-0.5 at various enlargements.

**Figure 5 polymers-15-00119-f005:**
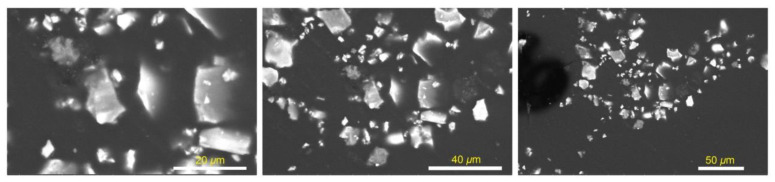
SEM images of PCP-16 at various enlargements.

**Figure 6 polymers-15-00119-f006:**
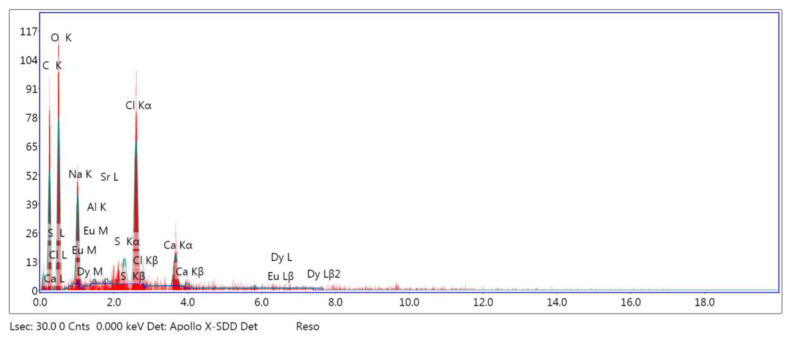
EDX spectra of PCP-4.

**Figure 7 polymers-15-00119-f007:**
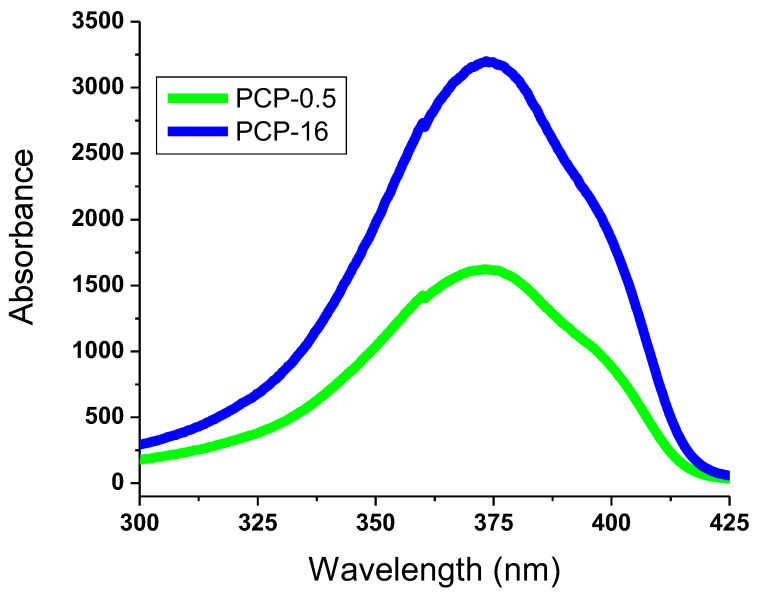
Excitation spectra of SAPN@PCP sheets.

**Figure 8 polymers-15-00119-f008:**
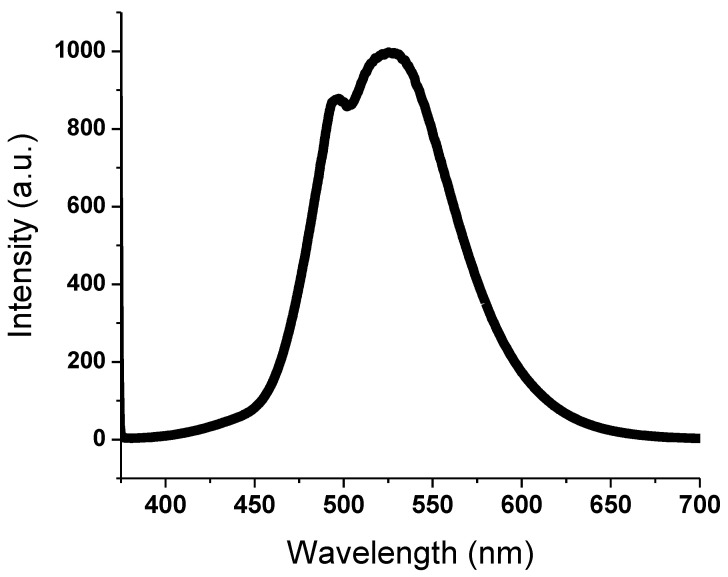
Emission spectra of SAPN@PCP sheets.

**Figure 9 polymers-15-00119-f009:**
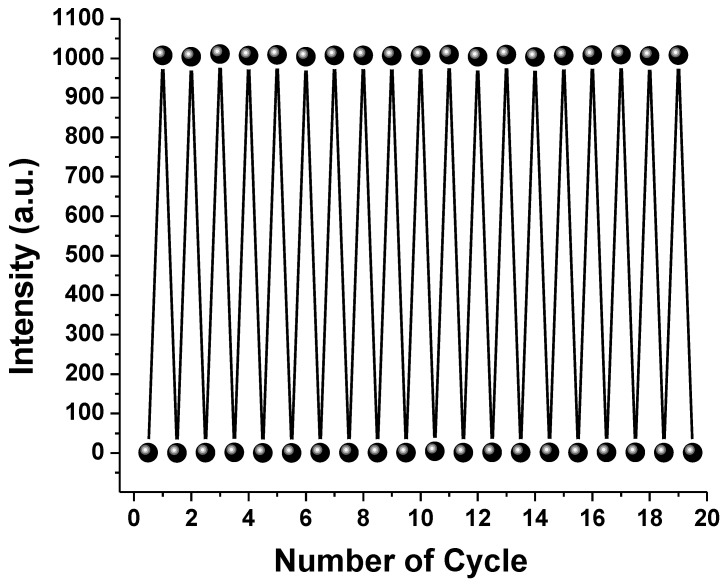
Fatigue resistance of PCP-12 (526 nm).

**Figure 10 polymers-15-00119-f010:**
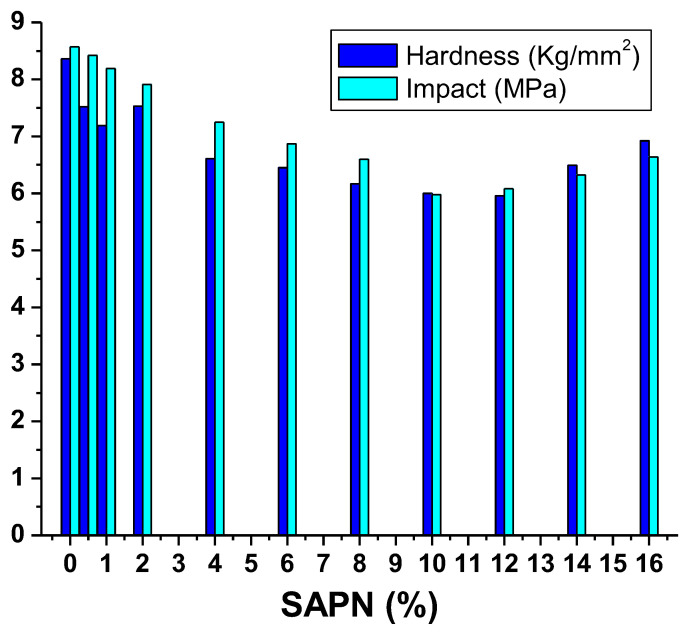
Mechanical efficiency of SAPN@PCP.

**Table 1 polymers-15-00119-t001:** Elements (wt%) monitored by EDX of SAPN@PCP at three separate points (P_1_, P_2_ and P_3_).

SAPN@PCP		C	O	Al	Sr	Eu	Dy
PCP-0	P_1_	65.82	34.18	0	0	0	0
	P_2_	66.24	33.76	0	0	0	0
	P_3_	66.19	33.81	0	0	0	0
PCP-0.5	P_1_	64.00	33.74	1.07	0.69	0.34	0.14
	P_2_	64.05	33.80	1.12	0.66	0.26	0.11
	P_3_	64.10	33.80	1.09	0.65	0.22	0.14
PCP-4	P_1_	61.11	33.59	2.57	1.67	0.72	0.34
	P_2_	60.81	33.81	2.64	1.72	0.59	0.43
	P_3_	60.75	33.76	2.76	1.80	0.61	0.32
PCP-8	P_1_	58.23	33.51	3.78	3.00	0.94	0.50
	P_2_	58.33	33.67	3.75	2.68	0.92	0.65
	P_3_	58.74	33.38	3.74	2.50	0.95	0.69
PCP-12	P_1_	56.20	32.13	5.50	4.34	1.06	0.82
	P_2_	56.11	32.03	5.83	4.10	1.17	0.76
	P_3_	56.43	32.32	5.30	3.99	1.34	0.73
PCP-16	P_1_	53.10	31.93	7.26	5.44	1.47	0.93
	P_2_	53.20	31.62	7.62	5.21	1.56	0.89
	P_3_	53.38	31.93	7.29	5.05	1.61	0.84

**Table 2 polymers-15-00119-t002:** Elements monitored by XRF in SAPN@PCP.

Element	Elemental Contents (wt%)				
	PCP-0.5	PCP-4	PCP-8	PCP-12	PCP-16
Na	2.60	2.48	1.63	1.38	0.84
K	2.55	2.26	1.85	0.71	0.77
Sr	25.72	27.88	30.33	32.17	33.06
Ca	3.06	2.48	1.82	1.08	0.80
Cl	3.39	2.37	1.41	0.96	0.75
Al	53.98	56.29	58.17	60.31	60.71
Si	8.70	6.14	4.79	2.39	2.07

**Table 3 polymers-15-00119-t003:** UPF and contact angle of SAPN@PCP.

SAPN@PCP	UPF	Contact Angle (°)
PCP-0	75	134.0
PCP-0.5	90	137.6
PCP-1	117	138.1
PCP-2	136	140.1
PCP-4	180	142.4
PCP-6	200	145.7
PCP-8	246	147.5
PCP-10	299	148.3
PCP-12	322	149.1
PCP-14	331	150.2
PCP-16	336	150.7

## Data Availability

Data Available from the corresponding author upon request.

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
