# Peer review of "Immobilization of Strontium Aluminate into Recycled Polycarbonate Plastics towards an Afterglow and Photochromic Smart Window"

_polymers, 2022, doi:10.3390/polym15010119_

Round 1
Reviewer 1 Report
In this report, the authors designed a kind of transparent smart window using the recycled polycarbonate as matrix, which was integrated with various amounts of SAPN. The research is very interesting, because PCP wastes are extremely difficult to be degraded, and transforming them into smart materials, like the smart window here is a target that most scientists are currently perusing. The researches are fine, and all the characterizations can support the conclusion. I suggest its publication on Polymers, but some issues should be addressed also.
1. What kind of PCP wastes had been used in this report, the author should specify it. Are they needed to be specially pretreated before mixing with SAPN? or use directly? What is the molecular weight of such PCP? What the PCP had been used before? All these information should be provided.
2. How did the author prepare the sample for SEM? Crack it in liquid N2 to see the cross section? The author should give the detailed descriptions.
3. Figure 3 and Figure 4 provided the dispersion information of 0.5% w/w and 16% w/w SAPN in PCP, however, for the two figures at a scale of 50 μm, it seems that the PCP-0.5 contained more amounts of SAPN. The author should explain it, or gave an another figure having bigger scale.
4. For SAPN, its size was claimed to be 15-52 nm by the author. However, after being dispersed in PCP, its size changed to 5-10 μm (calculated from the left figure of PCP-16). Did the morphology or size of SAPN change during the manufacture of PCP films? Because of what reasons?
Author Response
Point-by-point response to Reviewers’ comments
Reviewer # 1:
Comments and Suggestions for Authors
In this report, the authors designed a kind of transparent smart window using the recycled polycarbonate as matrix, which was integrated with various amounts of SAPN. The research is very interesting, because PCP wastes are extremely difficult to be degraded, and transforming them into smart materials, like the smart window here is a target that most scientists are currently perusing. The researches are fine, and all the characterizations can support the conclusion. I suggest its publication on Polymers, but some issues should be addressed also.
- What kind of PCP wastes had been used in this report, the author should specify it. Are they needed to be specially pretreated before mixing with SAPN? or use directly? What is the molecular weight of such PCP? What the PCP had been used before? All these information should be provided.
Response: Thanks for the your recommendations. Recycled polycarbonate plastic (PCP) were obtained from the Egyptian local marketplace (Aquahgroup). Transparent polycarbonate from plastic bottles was employed in the current study. The molecular weight of polycarbonate plastic bottles typically ranges between 50k – 300k g.mol-1. In order to prepare polycarbonate concrete bars, the PCP waste was firstly shredded, and then placed into a hot bath (110 °C) to remove the colored substrates to provide a clear plastic.
- How did the author prepare the sample for SEM? Crack it in liquid N2 to see the cross section? The author should give the detailed descriptions.
Response: Thanks for your recommendation. The sample was explored without further steps or coating with gold. The cross-section study showed no differences.
- Figure 3 and Figure 4 provided the dispersion information of 0.5% w/w and 16% w/w SAPN in PCP, however, for the two figures at a scale of 50 μm, it seems that the PCP-0.5 contained more amounts of SAPN. The author should explain it, or gave an another figure having bigger scale.
Response: Thanks for your feedback. The presence of SAPN on the sample surface does not reflect the exact amount of SAPN in the plastic bulk. The authors tried to select a good area on the samples surfaces to explore the presence of SAPN.
- For SAPN, its size was claimed to be 15-52 nm by the author. However, after being dispersed in PCP, its size changed to 5-10 μm (calculated from the left figure of PCP-16). Did the morphology or size of SAPN change during the manufacture of PCP films? Because of what reasons?
Response: Thanks for your recommendations. The size of SAPN was claimed to be 15-52 nm according to TEM study. The bigger size of SAPN on the sample surface could be attributed to the aggregation of particles during the molding process according to references bellow.
- Salhah D. Al-Qahtani, Adel M. Binyaseen, Enas Aljuhani, Meshari Aljohani, Hanan K. Alzahrani, Reem Shah, and Nashwa M. El-Metwaly. Ceramics International 48, no. 1 (2022): 903-912.
- Tawfik A. Khattab, Mahmoud Abd El‐Aziz, Meram S. Abdelrahman, Manal El‐Zawahry, and Samir Kamel. Luminescence 35, no. 4 (2020): 478-485.

Reviewer 2 Report
Manuscript is written well; the quality of figures is acceptable, and they are supported with sufficient discussions. I would like to recommend this manuscript for publication in Polymers after responding to below comments:
1- Abstract should be shorter (typically around 200-220 words). Please only highlight the principal concepts of your study and the main achievements. No need to mention all the characterization techniques and other minor points.
2- Please present the XRD and SAED patterns (in Figure 1) to show the crystallinity of SAPN materials.
3- Please present the EDX spectra and mapping analysis of materials using either SEM or TEM in addition to Table 1.
4- SEM images (Figures 3 and 4) should be replaced with clear images The dark background should be minimized to reach a satisfying resolution and quality. It’s impossible to get any useful information from the current SEM images.
5- Figure 6: Why the unit of Intensity (vertical axis) is nm? Authors should correct it to “Intensity (Counts)”.
6- Title is too wordy and can be shorter for better clarity.
7- Abbreviations should be defined at first mention and used consistently thereafter. Please remove redundant abbreviations and acronyms throughout the text.
8- EDX is a generally-accepted abbreviation for energy-dispersion X-ray analysis.
9- Figure 8 should possess suitable labels for the vertical axis.
10- References should be carefully rechecked and completed. Volume/page/paper numbers are missed in many refs.
Author Response
Point-by-point response to Reviewers’ comments
Reviewer # 2:
Comments and Suggestions for Authors
Manuscript is written well; the quality of figures is acceptable, and they are supported with sufficient discussions. I would like to recommend this manuscript for publication in Polymers after responding to below comments:
1- Abstract should be shorter (typically around 200-220 words). Please only highlight the principal concepts of your study and the main achievements. No need to mention all the characterization techniques and other minor points.
Response: Thanks for your recommendation. Abstract was shortened to only highlight the principal concepts and the main achievements of the current study, please see the revised manuscript.
2- Please present the XRD and SAED patterns (in Figure 1) to show the crystallinity of SAPN materials.
Response: Thanks for your comment. Figure 2 displays the TEM images (a-c), and selected area (electron) diffraction (d) of SAPN. XRD pattern of SAPN is shown in Figure 3. The diffraction peaks of SAPN were identical to the pure monoclinic phase of SrAl2O4 [24]. No other diffraction peaks were detected to indicate a complete integration of Eu2+ and Dy3+ in the SrAl2O4 crystal lattice [24], and to prove low temperature monoclinic phase content.
3- Please present the EDX spectra and mapping analysis of materials using either SEM or TEM in addition to Table 1.
Response: Thanks for your comment. Figure 6 displays EDX spectra of PCP-4. The extra elements shown in the EDX spectra could be attributed to contaminants accumulated on the sample surface with time.
4- SEM images (Figures 3 and 4) should be replaced with clear images The dark background should be minimized to reach a satisfying resolution and quality. It’s impossible to get any useful information from the current SEM images.
Response: We have re-studied the SEM analysis; however, we obtained the same results. The size of SAPN was claimed to be 15-52 nm according to TEM study. The bigger size of SAPN on the sample surface could be attributed to the aggregation of particles during the molding process according to references bellow.
- Salhah D. Al-Qahtani, Adel M. Binyaseen, Enas Aljuhani, Meshari Aljohani, Hanan K. Alzahrani, Reem Shah, and Nashwa M. El-Metwaly. Ceramics International 48, no. 1 (2022): 903-912.
- Tawfik A. Khattab, Mahmoud Abd El‐Aziz, Meram S. Abdelrahman, Manal El‐Zawahry, and Samir Kamel. Luminescence 35, no. 4 (2020): 478-485.
5- Figure 6: Why the unit of Intensity (vertical axis) is nm? Authors should correct it to “Intensity (Counts)”.
Response: Thanks for your comment. The unit of Intensity (vertical axis) was corrected to “Intensity (a.u.)”.
6- Title is too wordy and can be shorter for better clarity.
Response: Thanks for your recommendation. Title was shortened for better clarity.
7- Abbreviations should be defined at first mention and used consistently thereafter. Please remove redundant abbreviations and acronyms throughout the text.
Response: Thanks for your recommendation. Abbreviations were defined at first mention and used consistently thereafter.
8- EDX is a generally-accepted abbreviation for energy-dispersion X-ray analysis.
Response: Thanks for your recommendation. EDX was used as an abbreviation for energy-dispersion X-ray analysis.
9- Figure 8 should possess suitable labels for the vertical axis.
Response: Figure 10 (previously Figure 8) describes the mechanical efficiency of SAPN@PCP, displaying two different factors (hardness in Kg/mm2; and impact in MPa) that were already reported in the figure.
10- References should be carefully rechecked and completed. Volume/page/paper numbers are missed in many refs.
Response: Thanks for your comment. References were carefully rechecked and completed.
Thank you.

Round 2
Reviewer 2 Report
Authors properly responded my comments. Manuscript can be accepted in the current form. Two minor points should be performed in advance:
1- Figure 2d: please index the SAED pattern and specify the crystallographic planes inside the rings/spots.
2- Figure 3: please replot the XRD pattern using origin or excel and show the standard positions of peaks from the relevant ICDD card below the specific XRD peaks. It's totally unclear the current presentation for the future readers.
Author Response
Point-by-point response to Reviewers’ comments Authors properly responded my comments. Manuscript can be accepted in the current form. Two minor points should be performed in advance:
1- Figure 2d: please index the SAED pattern and specify the crystallographic planes inside the rings/spots.
Response: Thanks for the reviewer recommendation. Figure 2d shows the HRTEM image taken from the edge of lanthanide-doped strontium aluminate particle. No visible dis-locations and defects were detected. The spacing of the adjacent lattice planes is ca. 4.45 Å, which is consistent with the interplanar spacing of (0 1 1) plane of monoclinic SrAl2O4 [24].
2- Figure 3: please replot the XRD pattern using origin or excel and show the standard positions of peaks from the relevant ICDD card below the specific XRD peaks. It's totally unclear the current presentation for the future readers.
Response: Thanks for the reviewer recommendation. The XRD pattern was re-plotted to show the standard positions of peaks from the relevant ICDD card below the specific XRD peaks. XRD pattern of SAPN and its standard spectrum are shown in Figure 3. The diffraction peaks of SAPN were identical to the pure monoclinic phase of SrAl2O4 (JCPDS card No. 74-0794) [24]. No other diffraction peaks were detected to indicate a complete integration of Eu2+ and Dy3+ in the SrAl2O4 crystal lattice [24], and to prove low temperature monoclinic phase content. The small contents of the lanthanide doping elements displayed almost no effects on the SrAl2O4 phase composition. Thank you.
